# Preference Distillation via Value based Reinforcement Learning

**Minchan Kwon**[1]     **Junwon Ko**[1]     **Kangil Kim**[2*]     **Junmo Kim**[1*]

[1]Korea Advanced Institute of Science and Technology (KAIST)
[2]Gwangju Institute of Science and Technology (GIST)

{kmc0207, kojunewon,junmo.kim}@kaist.ac.kr, kikim01@gist.ac.kr,

## Abstract

Direct Preference Optimization (DPO) is a powerful paradigm to align language models with human preferences using pairwise comparisons. However, its binary win-or-loss supervision often proves insufficient for training small models with limited capacity. Prior works attempt to distill information from large teacher models using behavior cloning or KL divergence. These methods often focus on mimicking current behavior and overlook distilling reward modeling. To address this issue, we propose *Teacher Value-based Knowledge Distillation* (TVKD), which introduces an auxiliary reward from the value function of the teacher model to provide a soft guide. This auxiliary reward is formulated to satisfy potential-based reward shaping, ensuring that the global reward structure and optimal policy of DPO are preserved. TVKD can be integrated into the standard DPO training framework and does not require additional rollouts. Our experimental results show that TVKD consistently improves performance across various benchmarks and model sizes.

## 1   Introduction

Direct Preference Optimization (DPO) [24] is a powerful paradigm for aligning large language models (LLMs) with human preferences. Unlike Reinforcement Learning with Human Feedback (RLHF) [22], DPO enables efficient learning through pairwise comparisons, without requiring additional rollouts and reward model training. However, since DPO supervision is provided only in terms of binary win-or-lose preferences over entire sequences, it becomes difficult to evaluate how good a given response is. This coarse-grained feedback offers limited insight into the magnitude or quality of preference, making it challenging to assess fine-grained improvements or to interpret alignment quality beyond simple pairwise comparison. In small language models (SLMs) which have limited capacity and language understanding, this issue becomes more crucial  [9, 16].

To address this issue, several methods have attempted to leverage the rich information from large teacher models through diverse ways such as KL-divergence [9, 16] or online rollouts [30, 2]. However, most prior works focus on imitating the behavior of the teacher, without modeling its long-term reward structure. This issue has been widely discussed in the inverse RL [1, 33, 21] literature, where it is known that mimicking behavior without reward modeling can lead to compounding errors and poor generalization. Moreover, prior methods often disregard the potential conflict between the reinforcement learning (RL) objective and teacher information, leading to misalignment with the intended preference signal in datasets. For example, simply integrating KL-divergence with the

---

*Co-supervising authors.

39th Conference on Neural Information Processing Systems (NeurIPS 2025).

teacher in the RL objective may result in conflict between following the teacher policy and optimizing for human preferences in the offline dataset. This undermines the original intent of DPO, which is to align policies with human preferences expressed in data.

In this work, we propose *Teacher Value-based Knowledge Distillation* (TVKD), a method that integrates soft reward labels from the teacher without interfering with the DPO reward structure and its global optimal policy. The key idea is to introduce a novel auxiliary reward term that leverages the value function of the teacher. These value functions provide estimates of the value function that serves as a proxy for the internal reward modeling of the teacher model. We add this value function to the RL objective in a form that satisfies Potential-based Reward shaping (PBRS) [20]. This formula theoretically guarantees that the comparison of the action-level Q-functions will not change, maintaining the optimal policy. We introduce these value functions as a soft sequence-level reward, enabling DPO to account not just for which sequence wins, but by how much. This adds more informative supervision to the originally binary feedback. Practically, it can be integrated into the DPO loss with minimal modification, providing a simple yet effective way to incorporate internal signals from the teacher model.

We demonstrate that TVKD achieves strong performance across multiple benchmarks for preference distillation, including MT-Bench [31], AlpacaEval [6], and the Open LLM Leaderboard [8]. Notably, our method requires no additional rollouts and only leverages teacher outputs on an existing offline DPO dataset, making it compatible with the standard training frameworks of DPO. In addition, we demonstrate that TVKD performs consistently across student models ranging from 0.5B to 3B parameters, and remains robust under a wide range of ablation.

Our main contributions are summarized as follows:

- We introduce Teacher Value-based Knowledge Distillation, a preference distillation method that leverages the value function of the teacher as a proxy for its reward model.

- We present a form of auxiliary reward term that satisfies potential-based reward shaping, suggesting a method for adding teacher information without conflict with the original DPO reward structure.

- We empirically validate TVKD on multiple preference benchmarks, demonstrating consistent improvements across various student model sizes and datasets.

## 2   Related Work

**Preference Optimization**   Preference optimization is a training framework for aligning LLM outputs with human preferences. The standard approach, Reinforcement Learning from Human Feedback [22], involves training a reward model and fine-tuning the policy using policy gradient methods such as proximal policy optimization [26]. However, due to the high cost and instability of these online settings, offline alternatives have become popular in recent work. DPO [24] eliminates the reward modeling phase by directly optimizing the log-ratio margin between preferred and dispreferred responses. Extensions such as SimPO[19] and WPO [32] further address issues such as length or sampling bias. To overcome the limitations of relying on sequence-level preference signals, recent work has proposed incorporating token-level supervision into preference optimization [17, 29]. In particular, [25] shows that DPO-trained models implicitly learn a soft Q-function over token-level actions, revealing a potential path for finer-grained alignment.

**Preference Distillation**   Preference distillation is a technique that uses knowledge distillation (KD) [12] for preference optimization. Preference distillation can be categorized into online and offline settings. In online settings, rollouts from either the teacher or student model can be used to guide training. PaLD [30] uses DPO with teacher responses as the selected answers and student responses as rejected answers. DPKD [16] use in the same setting from PaLD, but replaces the reference term of DPO with the KL-divergence of teacher model, encouraging the student to follow the teacher closely. ADPA [9] leverages the difference between DPO-trained and SFT-trained teachers to generate feedback from student rollouts, which is used to train the student via policy gradient. While effective, such online methods often suffer from computational overhead and stability issues due to sampling during training. Offline preference distillation offers the advantage of eliminating rollouts during training, as it learns from a pre-collected dataset of human preferences. DCKD [9]

minimizes KL divergence between student and teacher outputs on both preferred and dispreferred responses. DDPO [7] makes the reward margin equalize between the student and teacher, which changes the RL objective to squared form. However, the offline setting has conflicts between the internal dataset reward and teacher reward, and stability concerns because it learns in an off-policy environment. We propose to solve this problem by using the value function of the teacher as a soft target for the first time to the best of our knowledge

**Reward Shaping in Reinforcement Learning** Reward shaping is a widely studied technique in RL for accelerating training and improving sample efficiency by providing additional, informative feedback beyond sparse environmental rewards. Potential-based reward shaping, introduced by Ng et al. [20], offers a theoretical guarantee that shaping terms can guide learning without altering the optimal policy. Subsequent work extended this idea to various RL settings, including temporal difference methods [28], partially observable MDPs [5], and multi-agent coordination [4]. We adapt this idea of potential-based shaping to preference distillation for language models.

## 3 Method

In this section, we introduce our method, Teacher Value-based Knowledge Distillation (TVKD). We first define language generation as a token-level Markov Decision Process (MDP), and re-visit value-based RL and DPO. We then define a value function from the DPO-trained teacher model, and incorporate it into the RL objective. We present a derivation of our final loss from the RL objective, along with an analysis of its behavior.

### 3.1 Preliminary

**Token-level MDP for Large Language Model** We formulate the language generation task as a token-level Markov Decision Process (MDP). Unlike the view of text generation as a static decision problem (e.g., multi-armed bandit), this formulation allows us to model the token-wise policies and their optimization over trajectories. Formally, We define the token-level MDP as a tuple $\mathcal{M} = (\mathcal{S}, \mathcal{A}, f, r, \rho_0)$, where the state space $\mathcal{S}$ consists of all tokens generated so far and the action space $\mathcal{A}$ is the vocabulary $\mathcal{V}$ of tokens. The dynamics $f$ are the deterministic transition model between the tokens $f(\mathbf{s}, \mathbf{a}) = \mathbf{s}|\mathbf{a}$, where $|$ is concatenation. The initial state distribution $\rho_0$ is a distribution over prompts $\mathbf{x}$, where an initial state $\mathbf{s}_0$ is comprised of the tokens from $\mathbf{x}$. The reward function $r(s, a)$ assigns a scalar reward to each state-action pair. This MDP structure provides a natural way to analyze and improve the generation policy $\pi_\theta$ over sequences by optimizing token-level decisions. Following previous token-level MDP settings, [24, 25] we fix the discount factor $\gamma = 1$.

**Value-based RL** To find a policy that maximizes the reward in a given token-level MDP, we can use the following objective:

$$\max_{\pi} \; \mathbb{E}_{\tau \sim \pi} \left[ \sum_{t=0}^{T} r(s_t, a_t) + \beta \, \mathcal{H}(\pi(\cdot|s_t)) \right],$$

where $\mathcal{H}(\pi(\cdot|s_t)) = -\sum_a \pi(a|s_t) \log \pi(a|s_t)$ is the Shannon entropy of the policy and $\beta$ balances the reward and exploration. This objective is called the Maximum Entropy RL (MaxEnt RL) objective.

Value-based RL offers a framework for solving MaxEnt RL problems by learning value functions that estimate expected future rewards. These functions act as surrogates for long-term returns and enable efficient policy improvement through dynamic programming or bootstrapping.

In this context, the **state value function** $V^\pi(s)$ is defined as the expected cumulative reward when starting from state $s$ and following policy $\pi$:

$$V^\pi(s) = \mathbb{E}_\pi \left[ \sum_{t=0}^{\infty} \gamma^t \, r(s_t, a_t) \; \middle| \; s_0 = s \right],$$

where $\gamma \in [0, 1)$ is a discount factor. The **action value function** $Q^\pi(s, a)$ additionally conditions on the action:

$$Q^\pi(s, a) = r(s, a) + \gamma \mathbb{E}_{s'} \left[ V^\pi(s') \right].$$

In the MaxEnt RL framework, the optimal state value function satisfies the soft Bellman consistency condition [11] as follows:

$$V^*(s) = \log \sum_a \exp\left(Q^*(s,a)\right), \tag{1}$$

which corresponds to the partition function over the exponentiated Q-values, where $Q^*$ and $V^*$ is optimal soft Q and value functions.

The optimal policy $\pi$ in value-based RL can be represented by Boltzman distribution of $Q^*$ and $V^*$ functions :

$$\pi^*(a \mid s) = \frac{\exp(Q^*(s,a))}{\sum_{a'} \exp(Q^*(s,a'))} = \exp\left(Q^*(s,a) - V^*(s)\right), \tag{2}$$

where the normalization constant is given by the soft value function $\exp(V^*(s)) = \sum_a \exp(Q^*(s,a))$. This form ensures that actions with higher Q-values are selected more frequently, while preserving stochasticity for exploration.

**Direct Preference Optimization**  While value-based RL are appealing because they explicitly model future returns, they are difficult to apply in practice without an online environment or an explicit reward model. Direct Preference Optimization (DPO) [24] offers a practical alternative by directly optimizing the policy from human preference data without requiring an explicit reward model. In typical preference learning settings, data are collected in the form of prompt-response comparisons $\mathcal{D} = \{(\mathbf{x}_i, \mathbf{y}_i^w, \mathbf{y}_i^l)\}^N$, where $\mathbf{y}^w$ is the preferred (or "winning") response and $\mathbf{y}^l$ is the less preferred (or "losing") response for the same prompt $\mathbf{x}$.

Each response is interpreted as a trajectory composed of token-level decisions. The Bradley–Terry model is used to express the probability that the winning trajectory is better than the losing one:

$$p^*(\tau^w \succ \tau^l) = \frac{\exp\left(r(\tau^w)\right)}{\exp\left(r(\tau^w)\right) + \exp\left(r(\tau^l)\right)},$$

where $r(\tau) = \sum_t r(s_t, a_t)$ is the total reward along a trajectory.

Rather than learning $r(s,a)$ directly, DPO derives a closed-form objective that allows preference likelihood to be optimized without explicit reward model training. By reparameterizing the reward in terms of the log-ratio between the learned policy $\pi_\theta$ and a reference policy $\pi_{\text{ref}}$, the resulting loss takes the following form:

$$\mathcal{L}_{\text{DPO}}(\pi_\theta, \mathcal{D}) = -\mathbb{E}_{(\tau^w, \tau^l) \sim \mathcal{D}} \left[ \log \sigma \left( \beta \sum_t \log \frac{\pi_\theta(a_t^w | s_t^w)}{\pi_{\text{ref}}(a_t^w | s_t^w)} - \beta \sum_t \log \frac{\pi_\theta(a_t^l | s_t^l)}{\pi_{\text{ref}}(a_t^l | s_t^l)} \right) \right],$$

where $\sigma(\cdot)$ denotes the sigmoid function and $\beta$ controls the scale of preference sensitivity.

### 3.2  Teacher Value-based Knowledge Distillation (TVKD)

**Overview**  We consider a setting where a DPO-style preference dataset $\mathcal{D}$ and a large, well-aligned teacher model $\pi_\phi$ are available. The teacher has been trained using DPO. Our goal is to train a smaller student model $\pi_\theta$ on $D$, leveraging sequence-level guidance from the teacher to improve alignment and generation quality.

**Soft Value Function of a DPO-trained Teacher**  DPO supervision is limited to binary sequence-level comparisons, offering no information about why a response is preferred or how much better it is. This lack of granularity makes it difficult to guide the model toward generating more aligned content. To address this, we propose to use the value function of the teacher model, which reflects the internal reward model.

Following the interpretation by Rafailov et al. [25], we treat a DPO-trained policy as a soft-optimal solution to a token-level MDP. In this framework, the token-level logits $Q_\phi(s,a)$ are viewed as soft Q-values, and the corresponding soft value function is defined according to MaxEnt RL principles.

**Lemma 1 (Soft value function of a DPO-trained policy [25]).**  *Let $Q_\phi(s,a)$ denote the token-level logits of a DPO-trained model $\pi_\phi$, and let $\beta > 0$ be the temperature of the Boltzmann policy. Then the soft value function is given by:*

$$V_\phi(s) = \beta \log \sum_{a \in \mathcal{V}} \exp(Q_\phi(s,a)/\beta).$$

In Appendix C.1, we provide full proof.

From a functional perspective, the soft value function can be interpreted as a soft measure of the number of high-quality choices available at a given state. It assigns higher values to states with many promising tokens, thereby indicating a more favorable decision point.

**Reward Shaping with Teacher Value**   We propose a method for incorporating information from the value function into the original DPO objective to provide sequence-level soft guidance. We define a shaping term based on the value function of the teacher:

$$\psi(s, a) = V_\phi(s') - V_\phi(s),$$

where $s' = f(s, a)$ denotes the next state. This shaping term captures the change in the expected future return and serves as a proxy for the utility of the action from the perspective of teacher.

Following the theoretical guarantee of Ng et al. [20], adding this value-based shaping term to the original DPO reward preserves the optimal policy:

**Lemma 2 (Optimal Policy Invariance under Teacher Value–based Shaping).**   *Let* $\mathcal{M} = (\mathcal{S}, \mathcal{A}, f, r, \rho_0)$ *be a token-level MDP, where* $r(\mathbf{s}, a)$ *denotes the implicit reward function derived from a dataset. Let* $\pi_\phi$ *be a DPO-trained teacher policy, and let* $V_\phi(\mathbf{s})$ *denote the state-value function of* $\pi_\phi$.

*Define the potential-based shaping function* $\psi(\mathbf{s}_t, a_t, \mathbf{s}_{t+1})$ *as:*

$$\psi(\mathbf{s}_t, a_t) = V_\phi(\mathbf{s}_{t+1}) - V_\phi(\mathbf{s}_t).$$

*Then, the optimal policies of the original MDP* $\mathcal{M}$ *and the shaped MDP* $\mathcal{M}' = (\mathcal{S}, \mathcal{A}, f, r + \psi, \rho_0)$ *are the same.*

*Proof Sketch (see Appendix C.2).*  By telescoping the shaping terms over a trajectory, the total shaped return differs from the original return by a state-dependent constant. As a result, the action that maximizes the expected return remains unchanged.  □

**Failure of Action-dependent Shaping**   In contrast to the value-based shaping term in Lemma 2, shaping functions that depend jointly on state and action, such as $\log \pi_\phi(a \mid s)$ or $Q_\phi(s, a)$, violate the sufficient conditions for policy invariance [20]. These action-dependent terms introduce implicit preferences over actions that cannot be factored out across trajectories, potentially distorting the underlying optimization.

**Corollary 2.1 (Action-based Shaping Breaks Policy Invariance).**   *Let* $r'(s, a) = r(s, a) + \psi(s, a)$, *where* $\psi(s, a)$ *is composed of the function dependent with both state and action. If* $\psi(s, a) \in \{\alpha \log \pi_\phi(a \mid s), \alpha Q_\phi(s, a)\}$, *then the resulting reward violates the policy invariance guarantee and may alter the optimal policy.*

We defer a formal proof of this corollary to Appendix C.3, where we show that action-dependent shaping terms fail to cancel out and lead to trajectory-level policy distortion.

**Teacher Value-based Knowledge Distillation**   Following Lemma 2, we incorporate teacher reward signals into student learning beyond binary supervision. This shaping term augments the original DPO reward, resulting in the following RL objective:

$$\max_{\pi_\theta} \mathbb{E}_{(s,a)\sim\pi_\theta} \left[ r(s, a) + \alpha \, \psi_\phi(s, a) \right],$$

where $\alpha$ is a hyperparameter that controls the strength of distillation.

Then we drive our final loss term from the new RL objective:

$$\mathcal{L}_{\text{TVKD}}(\pi_\theta, \mathcal{D}; \pi_\phi) = -\mathbb{E}_{(\tau_w, \tau_l)\sim\mathcal{D}} \left[ \log \sigma \left( \beta \log \frac{\pi_\theta(a^w \mid s^w)}{\exp\left(\frac{\alpha}{\beta}\psi_\phi(s^w, a^w)\right)} - \beta \log \frac{\pi_\theta(a^l \mid s^l)}{\exp\left(\frac{\alpha}{\beta}\psi_\phi(s^l, a^l)\right)} \right) \right].$$
$$(3)$$

We provide a full derivation in Appendix B.1. We have removed the MaxEnt RL term from the entire formula for simplicity, which can naturally be incorporated into the current formula. A full MaxEnt RL interpretation is in Appendix B.3.

**Gradient Analysis**   To further analyze how the TVKD loss function works, we define rewards over the preferred and less-preferred trajectories.

$$r_{\text{TVKD}}(s, a) := \beta \log \pi_\theta(a \mid s) - \alpha \, \psi_\phi(s, a),$$

where $V_\phi(s_T)$ is a terminal value of the value function of the teacher.

The total trajectory-level preference margin is computed as:

$$M := \sum_{t=0}^{|\tau_w|-1} r_{\text{TVKD}}(s^w, a^w) - \sum_{t=0}^{|\tau_l|-1} r_{\text{TVKD}}(s^l, a^l),$$

which accounts for the potential length mismatch between the preferred and less-preferred responses.

Then, the gradient of the TVKD loss becomes:

$$\nabla_\theta \mathcal{L}_{\text{TVKD}} = \sigma(-M) \cdot \left( \sum_{t=0}^{|\tau_w|-1} \nabla_\theta \log \pi_\theta(a_t^w \mid s_t^w) - \sum_{t=0}^{|\tau_l|-1} \nabla_\theta \log \pi_\theta(a_t^l \mid s_t^l) \right), \quad (4)$$

This expression shows that the gradient encourages increasing the log-probability of tokens in the preferred trajectory while decreasing that of tokens in the less-preferred one, modulated by the total preference margin $M$. Notably, the $M$ incorporates both student likelihoods and teacher-derived shaping, effectively blending supervised learning with policy shaping based on external value signals.

## 4   Experiments

### 4.1   Experiments Setting

**Backbone Architectures and Training Dataset**   We conduct the preference distillation experiments using three Small Language Models (SLM) as a student: LLaMA-3.2-1B, LLaMA-3.2-3B [10] and Danube3-500M [23]. For teacher models, we use Mistral-7B-v0.2 [13] and LLaMA-3.1-8B [10]. Following [9], we apply supervised fine-tuning (SFT) to both student and teacher models using Deita-10k-V0 [18], a dataset of 10k high-quality instruction-response pairs. For preference distillation, we use two datasets: (1) DPO-MIX-7K[2], a curated collection of high-quality pairwise preference data, and (2) HelpSteer2 [27], which is designed to improve helpfulness in LLMs. When using HelpSteer2, we exclude samples in which positive and negative responses have identical helpfulness scores.

**Validation and Evaluation Framework**   Following [9], we use Fsfair-LLaMA3-RM-V0.1 (RM)[3] to select the best checkpoints during training. RM generates a scalar score for the responses generated by the model for a given prompt, which has proven to perform well in RewardBench [15]. For evaluation, we evaluate our models on four benchmarks: RM, MT-Bench (MT) [31], AlpacaEval (AE) [6], and the Open LLM Leaderboard (OLL). For RM, we use the test set for evaluation, note that this is distinct from the validation set used for checkpoint selection. This gives the average score for the test prompts on the DPO-MIX-7K and Helpsteer2. For MT-Bench, we adopt the single-turn evaluation setup using the gpt-4o-mini-2024-07-18 model [4] as the evaluator. In AlpacaEval, we compute win rates against ADPA [9], also using gpt-4o-mini-2024-07-18 as the evaluator. For the Open LLM Leaderboard, we follow the evaluation protocol defined by the Gao et al. [8].

**Baselines**   We compare our method with diverse baselines, which can be broadly grouped into three categories. First, we consider KD methods. VanillaKD [12] and DCKD [9] are offline KD methods that rely solely on KL-divergence between the teacher and the student output. In contrast, SeqKD [14] and GKD [2] perform distillation in an online setting incorporating teacher or student rollouts, respectively. Second, we consider offline alignment methods. This group includes DPO[24] and its variants, SimPO [19] and WPO [32], which align model outputs with human preferences without teacher model information. TDPO [17] utilizes a token-level extension of DPO, which is from

---

[2]https://huggingface.co/datasets/argilla/dpo-mix-7k
[3]https://huggingface.co/sfairXC/FsfairX-LLaMA3-RM-v0.1
[4]https://platform.openai.com/docs/models/gpt-4o-mini

Table 1: Overall preference distillation performance. We use LLaMA-3.2-1B (student) and LLaMA-3.1-8B (teacher). We report the Fsfair-LLaMA3-RM-V0.1 (RM) rating, the alpaca eval win rate(AE) against ADPA, the average MT-bench ratings (MT), and Open LLM Leaderboard (OLL) scores. The best performances are highlighted in **bold**, while second-best performances are underline.

| Method | DPOMIX | | | | Helpsteer2 | | | |
|---|---|---|---|---|---|---|---|---|
| | RM(↑) | MT(↑) | AE(↑) | OLL(↑) | RM(↑) | MT(↑) | AE(↑) | OLL(↑) |
| DPO Teacher | -0.77 | 5.74 | 73.12 | 53.45 | -0.88 | 5.63 | 72.87 | 57.95 |
| SFT Teacher | -0.94 | 5.58 | 71.02 | 53.15 | -1.23 | 5.58 | 70.31 | 53.15 |
| SFT | -1.55 | 3.56 | 43.62 | 40.18 | -1.78 | 3.43 | 45.49 | 41.06 |
| DPO | -1.41 | 3.70 | 47.03 | 35.48 | -1.22 | 3.77 | 44.5 | 41.24 |
| SimPO | -1.22 | 3.65 | 46.12 | 41.5 | -1.37 | 3.75 | 49.98 | 41.33 |
| WPO | -1.51 | 3.53 | 39.23 | 41.36 | -1.62 | 3.62 | 48.35 | 41.19 |
| TDPO | -1.45 | 3.71 | 41.83 | 40.95 | -1.84 | 3.61 | 46.11 | 41.14 |
| VanillaKD | -1.66 | 3.46 | 38.88 | 41.17 | -1.70 | 3.53 | 44.06 | 41.09 |
| SeqKD | -1.38 | 3.40 | 37.16 | 41.17 | -1.75 | 3.20 | 38.47 | 41.17 |
| DDPO | -1.53 | 3.61 | 37.3 | 41.23 | -1.64 | 3.58 | 43.97 | 41.23 |
| DPKD | -1.57 | 3.43 | 37.99 | 41.35 | -1.73 | 3.48 | 43.04 | 41.14 |
| GKD | -1.61 | 3.52 | 36.63 | 41.19 | -1.74 | 3.41 | 40.78 | 40.92 |
| DCKD | -1.60 | 3.55 | 36.79 | 41.33 | -1.46 | 3.51 | 49.98 | 41.09 |
| ADPA | -1.21 | 3.73 | 50.00 | 40.29 | -1.43 | 3.57 | 50.00 | 41.26 |
| **TVKD(Ours)** | **-1.15** | **3.97** | **52.18** | **41.61** | **-1.18** | **3.98** | **54.85** | **41.35** |

**Question: How does photosynthesis work?**

| *DPO teacher* | *DPO student* | *TVKD student* |
|---|---|---|
| In photosynthesis, plants convert **sunlight**, water, and carbon dioxide into glucose and oxygen using **chlorophyll** to **capture** energy. | In photosynthesis, **plants convert** sunlight, water, and carbon dioxide into **glucose** and oxygen using chlorophyll to capture energy. | In photosynthesis, plants convert **sunlight**, water, and carbon dioxide into glucose and oxygen using **chlorophyll** to **capture** energy. |

Figure 1: Visualization of shaping terms. The top-3 tokens are highlighted in red.

self-judgment for token-level weight. Third, we consider preference distillation methods. DDPO [7] and DPKD [16] are fully offline approaches that do not require student rollouts. In contrast, ADPA [9] performs one round of student rollout over the entire dataset. More detailed training configurations can be found in Appendix E.1. We report the results we have reproduced.

## 4.2 Main Results

**Quantitative Results** We show the performance table when using LLaMA-3.1-8B as the teacher and LLaMA-3.2-1B as the student, training and evaluated on the DPOMIX and Helpsteer2 datasets in Table 1. First, our method consistently outperforms all baselines on both the RM and MT-Bench. These benchmarks evaluate response quality using LLM-based or GPT-based grading. Our method surpasses ADPA by 0.29% points on DPOMIX and 0.41% points on Helpsteer2 in MT-Bench. Notably, unlike ADPA, our approach requires no additional student rollouts or online evaluation by the teacher. This shows that TVKD has strong generation performance under a fully offline setting. In addition, compared to DDPO, which aligns the overall reward margin between the teacher and student, our method yields better performance. This suggests rather than simply mimicking the teacher's reward, our value function serves as a more effective in-

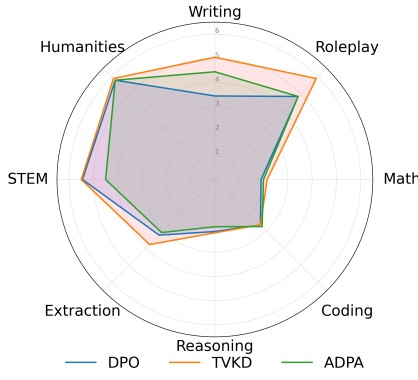

Figure 2: Visualization of task-wise performance on MT-bench.

Table 2: Results of ablation for TVKD using DPOMIX dataset. The best performances are highlighted in **bold**, while second-best performances are underline.

| | Mistral-7B->Danube-500M | | Mistral-7B -> Llama-1B | | Llama-8B->Llama-3B | |
| | RM($\uparrow$) | MT($\uparrow$) | RM($\uparrow$) | MT($\uparrow$) | RM($\uparrow$) | MT($\uparrow$) |
|---|---|---|---|---|---|---|
| DPO | -3.03 | 3.21 | -1.61 | 3.28 | -1.10 | 5.08 |
| SimPO | -2.91 | 2.96 | -1.55 | 3.32 | -1.12 | 4.90 |
| DCKD | -2.08 | 2.84 | -1.72 | 3.07 | -1.25 | 4.86 |
| Ours | **-1.99** | **3.22** | **-1.36** | **3.38** | **-1.05** | **5.19** |

Table 3: Accuracy of identifying preferred trajectories using different reward formulations, log probabilities (DPO), length-normalized log probabilities (SimPO), and our shaping term (Ours).

| Scoring term | Log prob. | Log prob. with LC | **Shaping term(Ours)** |
|---|---|---|---|
| Accuracy(%) ($\uparrow$) | 19.56 | 19.72 | **25.82** |

dicator of human preference. Second, our method achieves strong performance under pairwise comparison-based evaluation using AE. On both datasets, it is the only method that surpasses ADPA in win rate, highlighting its strength in direct pairwise comparisons beyond scalar scoring. This means that TVKD produced higher-quality sentences than the previous best model, ADPA, even in a relative evaluation that is more precise than a sentence-by-sentence evaluation. Finally, we show competitive performance on OLL benchmarks, demonstrating that our method generalizes well across a variety of domains. We report only the average results in the main table, while full results are provided in Appendix F.1. Our method outperforms all baselines in OLL, indicating that TVKD achieves strong alignment without sacrificing performance in other areas.

To further investigate the effect of value-based distillation, we analyze the performance of each task in MT-bench, as shown in Figure 2. We compare our method with DPO and ADPA, representing methods without and with distillation, respectively. Comparisons with other methods can be found in Appendix F.2. TVKD outperforms both baselines across all areas. In particular, our method outperforms DPO in Writing and Roleplay. A similar trend is observed in ADPA, indicating that distillation helps transfer the teacher's writing ability. On the other hand, compared to ADPA, TVKD shows strengths in STEM and Extraction. This is where DPO has an advantage over ADPA, which shows that TVKD takes the strengths of both DPO and ADPA.

**Qualitative Results**     Figure 1 visualizes the shaping term, defined as the difference in value functions between consecutive tokens. The top-3 tokens with the highest values are highlighted in red. For TVKD, which is trained using the value function of a DPO teacher, the highlighted tokens such as *sunlight*, *chlorophyll*, and *capture* match those selected by the teacher. These tokens are conceptually central to the explanation of photosynthesis, indicating that the value function effectively captures semantically meaningful tokens. While the DPO student shares the same training objective as the DPO teacher, its smaller model size leads to differences in the learned value function. This suggests that TVKD successfully inherits the teacher's reward modeling behavior.

**Model Ablation**     In Table 2, we evaluate TVKD across various teacher-student configurations on the DPOMIX dataset. For computational limitations, we compare only offline methods. TVKD consistently outperforms all baselines, demonstrating strong generalization across model scales and architectures. Notably, it improves performance by 0.3% points even when distilling from Mistral-7B into the compact Danube2.1-500M, showing that our value guidance remains effective for smaller models. TVKD also demonstrates robust performance across teacher architectures, confirming the transferability of teacher-derived value signals. Moreover, TVKD performs robustly when the size difference between the teacher and student model is smaller.

Table 4: Analysis of value function alignment and MT-Bench performance across models. We compare the KL-divergence between teacher and student value functions.

| Metric | DPO Teacher | DPO Student | **Ours** |
|---|---|---|---|
| KL-div. with Teacher Value ($\downarrow$) | 0.000 | 1.245 | **1.202** |
| MT-Bench ($\uparrow$) | 5.74 | 3.70 | **3.97** |

Table 5: Comparison of various auxiliary rewards on the same setting in Table 1. In Alpaca-Eval (AE), we use our method as a baseline.

| Type | Name | Auxiliary Reward | Margin Acc.($\uparrow$) | MT-bench($\uparrow$) | AE($\uparrow$) |
|---|---|---|---|---|---|
| Action Dependent | Logits | $Q(a, s)$ | 18.29 | 3.61 | 37.18 |
| | Log Probability | $\log \pi_\phi(a \mid s)$ | 18.31 | 3.43 | 32.50 |
| State Dependent | Max | $\max_a \pi_\phi(a \mid s)$ | 28.86 | 3.79 | 36.07 |
| | Margin | $\pi_\phi^{(1)}(s) - \pi_\phi^{(2)}(s)$ | 30.23 | 3.90 | 48.81 |
| | Expectation | $\sum_a \pi_\phi(a \mid s)Q_\phi(a \mid s)$ | 30.23 | 3.45 | 49.42 |
| | **Ours** | $\log \sum_a \exp\left(Q_\phi(a \mid s)\right)$ | 30.23 | **3.97** | **50.00** |

## 4.3 Analysis

**Value Function Analysis**   To investigate whether the value function is good at giving pairwise soft labels, we perform additional analysis in  Table 3.  Specifically, we compute sequence-level rewards by summing token-wise values derived from different signals: log-probabilities (DPO), log-probabilities with length normalization (SimPO), and shaping terms (Ours). Our shaping term achieves higher accuracy than both the raw and the length-normalized log probabilities, suggesting that the teacher value function provides a more accurate estimate of human preference. Because our method uses a telescoping sum, it naturally reduces the impact of sequence length without requiring explicit normalization. As a result, it performs better than methods that rely on length normalization. This shows that our shaping term can act as a reliable and consistent sequence-level soft label.

**Value Alignment Analysis**   In Table 4, we analyze whether TVKD is simulating the value function of the teacher well. We provide MT-bench performance and value function distance with DPO teacher for three models, DPO teacher, DPO student, and TVKD. For value function distance, it is measured for chosen responses and questions from the train set of DPOMIX and measures the KL-divergence between the output of the teacher value function and the target value function. TVKD shows lower KL-divergence for the teacher value function compared to DPO students. This shows that the student model trained with TVKD has a value function that is more similar to the DPO teacher. This shows that TVKD is implicitly modeling the teacher's reward modeling during training, even though it does not have an object that directly models the value function.

**Reward Conflict Analysis**   In Table 5 we conduct a detailed analysis to investigate the effects of PBRS and the design of auxiliary reward functions. Specifically, we evaluate various auxiliary reward functions intended to reflect the reward signal of the teacher. To check the conflict between the reward from the dataset and the auxiliary reward, we demonstrate margin accuracy. The margin accuracy is computed following the protocol in [19], measuring the proportion of test samples in which the chosen response receives a higher reward than the rejected one. Action-dependent functions, such as logits or log probabilities, fail to preserve the optimal policy theoretically (see Corollary 2.1). Empirically, they also result in lower margin accuracy, confirming their misalignment with preference data. In contrast, all state-dependent functions satisfy PBRS by design, but their performance varies depending on the quality of sequence-level supervision. In practice, all state-dependent functions exhibit high margin accuracy. However, our value function achieves the highest performance, showing that theoretical guarantees alone are insufficient. These results suggest that effective alignment requires accurate and semantically meaningful reward shaping.

## 5 Conclusion

In this paper, we tackled the challenge of aligning SLMs with human preferences, a task made difficult by the limited capacity of such models and the coarse granularity of sequence-level feedback. To address this, we introduced Teacher Value-based Knowledge Distillation, a novel offline distillation framework that interprets preference alignment through the lens of reward shaping. By extracting value functions from a preference-aligned teacher model, TVKD reshapes the DPO objective to enable fine-grained, trajectory-consistent supervision without altering the optimal policy. Extensive experiments across various benchmarks confirm that TVKD consistently enhances performance.

## Acknowledgments

This work was supported by Institute of Information communications Technology Planning Evaluation (IITP) grant funded by the Korea government (MSIT) (No.2022-0-00951, Development of Uncertainty-Aware Agents Learning by Asking Questions).

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

## A  Limitation

Although our method theoretically guarantees optimal policy invariance under value-based shaping [20], in practice, the shaping term can negatively affect training dynamics if the teacher's value function is severely misestimated or noisy. Although our method guarantees optimal policy invariance, it can be reach in suboptimal. Our method assumes that the teacher value function provides reliable signals. If the value is highly inaccurate or noisy, the shaping term may interfere with learning and lead to suboptimal policies. We do not test cases with weak or mismatched teachers, and the method's robustness in such settings is unclear.

## B  Mathematical Derivation

### B.1  Mathematical Derivation of Equation 3

We begin with the value-guided shaping framework introduced in our main objective:

$$\max_{\pi_\theta} \mathbb{E}_{\mathbf{a}_t \sim \pi_\theta(\cdot | \mathbf{s}_t)} \left[ \sum_{t=0}^{T-1} \left( r(\mathbf{s}_t, \mathbf{a}_t) + \alpha \, \psi_\phi(\mathbf{s}_t, \mathbf{a}_t) \right) \right], \tag{5}$$

where the shaping term is defined as:

$$\psi_\phi(s, a) := V_\phi(s') - V_\phi(s),$$

and $s'$ is the next state resulting from action $a$ in state $s$.

To connect this to the DPO setting, we follow the formulation in [25], which relates the reward $r(s, a)$ to a soft Q-function $Q^*(s, a)$. In our setting, the shaped Q-function takes the form:

$$Q^*(s_t, a_t) = r(s_t, a_t) + \psi_\phi(s_t, a_t) + \alpha \, V^*(s_{t+1}), \tag{6}$$

assuming $s_{t+1}$ is not terminal. At terminal states, $V^*(s_T) = 0$.

Using this, we express the total cumulative reward across a trajectory $\tau = (s_0, a_0, \ldots, a_{T-1}, s_T)$ as:

$$\sum_{t=0}^{T-1} r(s_t, a_t) = \sum_{t=0}^{T-1} \left( Q^*(s_t, a_t) - V^*(s_{t+1}) - \alpha \, \psi_\phi(s_t, a_t) \right)$$

$$= Q^*(s_0, a_0) + V^*(s_0) + \sum_{t=1}^{T-1} \left( Q^*(s_t, a_t) - V^*(s_t) - \alpha \, \psi_\phi(s_t, a_t) \right),$$

Applying the DPO interpretation of the soft Q-function as log-probabilities [25], we approximate: $Q^*(s_t, a_t) - V^*(s_{t+1}) = \beta \log \pi_\theta(a_t \mid s_t)$ and substitute $\psi_\phi(s_t, a_t) = V_\phi(s_{t+1}) - V_\phi(s_t)$ to obtain the shaped log-prob objective:

$$\sum_{t=0}^{T-1} r(s_t, a_t) = \sum_{t=0}^{T-1} \beta \log \frac{\pi_\theta(a_t \mid s_t)}{\exp\left( \frac{\alpha}{\beta} \psi_\phi(s_t, a_t) \right)} + V^*(s_0).$$

Thus, the TVKD loss over a pairwise preference dataset $\mathcal{D}$ becomes:

$$\mathcal{L}_{\text{TVKD}}(\pi_\theta, \mathcal{D}; \pi_\phi) = -\mathbb{E}_{(\tau_w, \tau_l) \sim \mathcal{D}} \left[ \log \sigma \left( \sum_{t=0}^{|\tau_w|-1} \beta \log \frac{\pi_\theta(a_t^w \mid s_t^w)}{\exp\left( \frac{\alpha}{\beta} \psi_\phi(s_t^w, a_t^w) \right)} \right. \right.$$

$$\left. \left. - \sum_{t=0}^{|\tau_l|-1} \beta \log \frac{\pi_\theta(a_t^l \mid s_t^l)}{\exp\left( \frac{\alpha}{\beta} \psi_\phi(s_t^l, a_t^l) \right)} \right) \right],$$

where $\psi_\phi(s, a) = V_\phi(s') - V_\phi(s)$.

Note that all conditions based on the initial state $s_0$ are shared between $\tau_w$ and $\tau_l$ and therefore are canceled out in the preference-based loss.

## B.2 Mathematical Derivation of the Equation 4

We begin by defining the *shaped reward* at each timestep $t$ using the teacher's potential-based shaping function $\psi_\phi$, which is independent of the student parameters $\theta$:

$$r_{\text{TVKD}}(a_t, s_t) = \beta \log \pi_\theta(a_t \mid s_t) - \alpha \psi_\phi(s_t, a_t).$$

For a pair of trajectories—one preferred ($w$) and one less preferred ($l$)—we define the total shaped reward as:

$$R^w := \sum_{t=0}^{|\tau_w|-1} r_{\text{TVKD}}(a_t^w, s_t^w), \quad R^l := \sum_{t=0}^{|\tau_l|-1} r_{\text{TVKD}}(a_t^l, s_t^l).$$

Then, the preference loss is given by the negative log-sigmoid of the total reward margin:

$$\mathcal{L}_{\text{TVKD}} = -\log \sigma(R^w - R^l),$$

where the scalar margin is:

$$M := R^w - R^l = \sum_{t=0}^{|\tau_w|-1} r_{\text{TVKD}}(a_t^w, s_t^w) - \sum_{t=0}^{|\tau_l|-1} r_{\text{TVKD}}(a_t^l, s_t^l).$$

Since $\psi_\phi$ is computed only from the teacher and does not depend on $\theta$, its gradient vanishes:

$$\nabla_\theta \psi_\phi(s_t, a_t) = 0.$$

Therefore, the gradient of the margin $M$ becomes:

$$\nabla_\theta M = \beta \left( \sum_{t=0}^{T^w-1} \nabla_\theta \log \pi_\theta(a_t^w \mid s_t^w) - \sum_{t=0}^{T^l-1} \nabla_\theta \log \pi_\theta(a_t^l \mid s_t^l) \right).$$

Applying the identity $\nabla_x \log \sigma(x) = \sigma(-x)$, the gradient of the loss becomes:

$$\nabla_\theta \mathcal{L}_{\text{TVKD}} = -\nabla_\theta \log \sigma(M) = \sigma(-M) \cdot \nabla_\theta M$$

$$= \beta \cdot \sigma(-M) \cdot \left( \sum_{t=0}^{T^w-1} \nabla_\theta \log \pi_\theta(a_t^w \mid s_t^w) - \sum_{t=0}^{T^l-1} \nabla_\theta \log \pi_\theta(a_t^l \mid s_t^l) \right).$$

This formulation generalizes to trajectories of unequal lengths and cleanly separates the teacher shaping from the student optimization path. The teacher's value function contributes to the reward margin, while the gradient flows solely through the student model's log-probabilities.

## B.3 Mathematical Derivation of Equation 3 in MaxEnt RL setting

We extend our value-guided objective to the entropy-regularized reinforcement learning (MaxEnt RL) framework. The resulting objective is:

$$\max_{\pi_\theta} \mathbb{E}_{\mathbf{a}_t \sim \pi_\theta(\cdot|\mathbf{s}_t)} \left[ \sum_{t=0}^{T} \left( r(\mathbf{s}_t, \mathbf{a}_t) - \mathbb{D}_{\text{KL}} \left[ \pi_\theta(\mathbf{a}_t|\mathbf{s}_t) \| \pi_{\text{ref}}(\mathbf{a}_t|\mathbf{s}_t) \right] + \alpha \, \psi_\phi(\mathbf{s}_t, \mathbf{a}_t) \right) \right], \quad (7)$$

where the shaping term is defined as $\psi_\phi(s, a) := V_\phi(s') - V_\phi(s)$, consistent with the formulation in our main objective.

We follow [25], which describes the recursive soft Q-function in token-level DPO as:

$$Q^*(s_t, a_t) = \begin{cases} r(s_t, a_t) + \beta \log \pi_{\text{ref}}(a_t|s_t) + V^*(s_{t+1}) + \alpha \, \psi_\phi(s_t, a_t), & \text{if } s_{t+1} \text{ is not terminal,} \\ r(s_t, a_t) + \beta \log \pi_{\text{ref}}(a_t|s_t) + V_\phi(s_t), & \text{if } s_{t+1} \text{ is terminal.} \end{cases}$$

$$(8)$$

Summing over a trajectory $\tau = (s_0, a_0, \ldots, a_{T-1}, s_T)$, the total reward becomes:

$$\sum_{t=0}^{T-1} r(s_t, a_t) = \sum_{t=0}^{T-1} [Q^*(s_t, a_t) - \beta \log \pi_{\text{ref}}(a_t|s_t) - V^*(s_{t+1}) - \alpha \, \psi_\phi(s_t, a_t)] \tag{9}$$

$$= \sum_{t=0}^{T-1} [\beta \log \pi_\theta(a_t|s_t) - \beta \log \pi_{\text{ref}}(a_t|s_t) - \alpha \, \psi_\phi(s_t, a_t)] \tag{10}$$

$$= \sum_{t=0}^{T-1} \beta \log \frac{\pi_\theta(a_t|s_t)}{\pi_{\text{ref}}(a_t|s_t)} - \sum_{t=0}^{T-1} \alpha \, \psi_\phi(s_t, a_t). \tag{11}$$

The final TVKD loss under MaxEnt RL:

$$\mathcal{L}_{TVKD}(\pi_\theta, \mathcal{D}; \pi_{ref}, \pi_\phi) = -\mathbb{E}_{(\tau_w, \tau_l) \sim \mathcal{D}} \Bigg[ \log \sigma \Bigg( \sum_{t=0}^{|\tau_w|-1} \beta \log \frac{\pi_\theta(a_t^w|s_t^w)}{\exp(\frac{\alpha}{\beta}\psi_\phi(s_t^w, a_t^w))\pi_{ref}(a_t^w|s_t^w)}$$

$$\tag{12}$$

$$- \sum_{t=0}^{|\tau_l|-1} \beta \log \frac{\pi_\theta(a_t^l|s_t^l)}{\exp(\frac{\alpha}{\beta}\psi_\phi(s_t^l, a_t^l))\pi_{ref}(a_t^l|s_t^l)} \Bigg) \Bigg]$$

## C  Proof of Lemma

### C.1  Proof of Lemma 1

We build on the interpretation introduced by Rafailov et al. [25], which characterizes DPO-trained policies as soft-optimal solutions in a deterministic token-level MDP. Specifically, they show that the token-level logits $Q_\phi(s, a)$ of a DPO-trained model correspond to the soft Q-function of an optimal Boltzmann policy under the maximum entropy reinforcement learning (MaxEnt RL) framework.

**Assumption (from [25]).**  Let $\pi_\phi(a \mid s)$ be the DPO-trained policy defined as:

$$\pi_\phi(a \mid s) = \frac{\exp(Q_\phi(s, a)/\beta)}{\sum_{a'} \exp(Q_\phi(s, a')/\beta)},$$

for a fixed temperature $\beta > 0$. Then $\pi_\phi$ is a soft-optimal policy with Q-function $Q_\phi(s, a)$, and its corresponding soft value function is defined by:

$$V_\phi(s) := \mathbb{E}_{a \sim \pi_\phi(\cdot|s)}[Q_\phi(s, a)] + \beta \mathcal{H}[\pi_\phi(\cdot \mid s)],$$

where $\mathcal{H}[\pi] = -\sum_a \pi(a) \log \pi(a)$ denotes the entropy of the policy.

We now derive a closed-form expression for $V_\phi(s)$ based on the Boltzmann structure of $\pi_\phi$.

**Derivation.**  First, recall the identity:

$$\log \pi_\phi(a \mid s) = \frac{Q_\phi(s, a)}{\beta} - \log \sum_{a'} \exp(Q_\phi(s, a')/\beta).$$

Thus, the entropy term becomes:

$$\mathcal{H}[\pi_\phi] = -\sum_a \pi_\phi(a \mid s) \log \pi_\phi(a \mid s)$$

$$= -\sum_a \pi_\phi(a \mid s) \left( \frac{Q_\phi(s, a)}{\beta} - \log \sum_{a'} \exp(Q_\phi(s, a')/\beta) \right)$$

$$= -\frac{1}{\beta} \sum_a \pi_\phi(a \mid s) Q_\phi(s, a) + \log \sum_{a'} \exp(Q_\phi(s, a')/\beta).$$

Now substitute back into the soft value function definition:

$$V_\phi(s) = \sum_a \pi_\phi(a \mid s) Q_\phi(s, a) + \beta \mathcal{H}[\pi_\phi]$$

$$= \sum_a \pi_\phi(a \mid s) Q_\phi(s, a) + \beta \left( -\frac{1}{\beta} \sum_a \pi_\phi(a \mid s) Q_\phi(s, a) + \log \sum_{a'} \exp(Q_\phi(s, a')/\beta) \right)$$

$$= \log \sum_{a \in \mathcal{V}} \exp(Q_\phi(s, a)/\beta).$$

This completes the proof.

## C.2 Proof of Lemma 2

In Section 3 we stated that augmenting the DPO reward with the Teacher Temporal Difference term $V_\phi(s_{t+1}) - V_\phi(s_t)$ does not change the optimal policy. Below we provide a formal proof.

*Proof.* Consider a trajectory $\tau = (s_0, a_0, s_1, a_1, \ldots, s_{T-1}, a_{T-1}, s_T)$. Let the original return be:

$$G(\tau) = \sum_{t=0}^{T-1} r(s_t, a_t).$$

Now, define a potential-based shaped reward:

$$\tilde{r}(s_t, a_t, s_{t+1}) = r(s_t, a_t) + \Phi(s_{t+1}) - \Phi(s_t),$$

and let the corresponding shaped return be:

$$\tilde{G}(\tau) = \sum_{t=0}^{T-1} [r(s_t, a_t) + \Phi(s_{t+1}) - \Phi(s_t)].$$

By telescoping,

$$\sum_{t=0}^{T-1} (\Phi(s_{t+1}) - \Phi(s_t)) = \Phi(s_T) - \Phi(s_0),$$

so the shaped return becomes:

$$\tilde{G}(\tau) = G(\tau) + \Phi(s_T) - \Phi(s_0).$$

This additive term depends only on the start and end states, not on the specific actions taken. Therefore, for any policy $\pi$, the expected shaped state-action value is:

$$\tilde{Q}^\pi(s, a) = Q^\pi(s, a) + \mathbb{E}_{s' \sim P(\cdot|s,a)}[\Phi(s')] - \Phi(s).$$

Since the difference between shaped Q-values at any state $s$ only depends on $\Phi(s)$ and the expected $\Phi(s')$, the ranking over actions remains unchanged:

$$\arg\max_a \tilde{Q}^*(s, a) = \arg\max_a Q^*(s, a).$$

Hence, the optimal policy is invariant under this form of reward shaping. $\square$

## C.3 Proof of Corollary 2.1

In Section 3 we claimed that action-based shaping terms such as $\psi(s, a) = \alpha \log \pi_\phi(a \mid s)$ or $\psi(s, a) = \alpha Q_\phi(s, a)$ violate the conditions required for potential-based shaping and may alter the optimal policy. We now provide a formal justification.

*Proof.* Suppose we augment the DPO reward with an action-dependent shaping term:

$$\tilde{r}(s_t, a_t) = r(s_t, a_t) + \psi(s_t, a_t), \quad \text{where} \quad \psi(s_t, a_t) \neq \Phi(s_t) - \Phi(s_{t+1}).$$

Then the shaped return for a trajectory $\tau = (s_0, a_0, s_1, \ldots, s_T)$ becomes:

$$\tilde{G}(\tau) = \sum_{t=0}^{T} \left[ r(s_t, a_t) + \psi(s_t, a_t) \right].$$

Unlike the state-based shaping case in Lemma 2, there is no telescoping cancellation of the added shaping terms because $\psi(s_t, a_t)$ depends explicitly on the action at each step and not solely on the state. As a result, the shaped return $\tilde{G}(\tau)$ differs from the original return $G(\tau)$ by an amount that depends on the entire sequence of actions:

$$\tilde{G}(\tau) = G(\tau) + \sum_{t=0}^{T} \psi(s_t, a_t),$$

where $\sum_t \psi(s_t, a_t)$ varies with the policy and trajectory.

This action-dependent distortion implies that:

$$\tilde{Q}^{\pi}(s, a) = Q^{\pi}(s, a) + \mathbb{E}_{\tau \sim \pi | (s_0 = s, a_0 = a)} \left[ \sum_{t=0}^{T} \psi(s_t, a_t) \right],$$

which can shift the relative ranking of actions at state $s$, thereby changing $\arg\max_a \tilde{Q}^*(s, a)$.

Hence, the optimal policy under $\tilde{r}$ may differ from the one under the original reward $r$, violating policy invariance. $\qquad\square$

## D    Relation to Advantage Function in RL

Our method can be interpreted as a special form of value-based reward shaping, closely related to the advantage function in reinforcement learning. In standard RL, the advantage function is defined as:

$$A(s, a) = r(s, a) + V(s') - V(s),$$

which quantifies the relative benefit of taking action $a$ at state $s$ compared to the expected return from the state baseline $V(s)$.

In our formulation, we do not use an explicit reward $r(s, a)$, as it is typically unavailable in offline preference datasets. Instead, we define the reward shaping term using the teacher's value function:

$$\psi(s, a) := V_\phi(s') - V_\phi(s),$$

where $V_\phi$ is the value function learned by the DPO teacher model, and $s'$ denotes the next state (i.e., the updated token-level context after applying action $a$).

This formulation implicitly captures a notion of advantage by measuring the change in future utility as estimated by the teacher. It provides directional feedback for the student model without relying on noisy or hard-to-estimate token-level rewards.

This connection offers a theoretical justification for the stability and effectiveness of our method, as value-based shaping of this form is known to preserve the optimal policy under standard conditions [20].

## E    Experiment Details

### E.1    Training Details

**Training Details**   The SFT for both student and teacher models is conducted over 3 epochs, using a learning rate of $2 \times 10^{-5}$ and a batch size of 128. The DPO teacher is trained with $\beta = 0.05$, a learning rate of $5 \times 10^{-7}$, a batch size of 128, and for 2 epochs. For distillation methods, we employ context distillation [3] to improve efficiency. Specifically, we precompute the top 50 probabilities and save them to use for distillation following [9]. In TVKD, we experiment with $\alpha \in [0.1, 0.2, 0.5, 0.7, 1.0, 1.5]$ and $\beta \in [0.1, 0.2, 0.5, 1, 2, 5]$.

Table 6: Overall performance of Open LLM Leaderboard trained with DPOMIX datasets.

| Method | Average | ARC-easy | GSM8K | Hellaswag | MMLU | TruthfulQA | Winogrande |
|---|---|---|---|---|---|---|---|
| DPO teacher | 53.448 | 76.800 | 47.900 | 61.480 | 60.230 | 04.200 | 70.080 |
| SFT teacher | 53.152 | 76.200 | 47.300 | 60.844 | 60.070 | 04.500 | 70.000 |
| SFT | 40.176 | 59.040 | 05.500 | 49.570 | 37.950 | 29.488 | 59.510 |
| DPO | 35.475 | 59.000 | 06.300 | 49.760 | 37.840 | 00.600 | 59.350 |
| SimPO | 41.497 | 64.942 | 06.975 | **50.129** | 37.943 | 29.009 | 59.984 |
| WPO | 41.356 | 64.850 | 06.369 | 49.642 | 37.958 | 29.253 | 60.063 |
| TDPO | 40.950 | 64.310 | 06.431 | 49.991 | 37.560 | 27.907 | 59.747 |
| VanillaKD | 41.166 | 64.980 | 05.838 | 49.353 | 37.922 | 28.764 | 60.141 |
| SeqKD | 41.166 | 64.980 | 06.141 | 48.840 | **37.820** | 29.002 | 60.210 |
| DDPO | 41.225 | 65.000 | 06.141 | 49.492 | 37.840 | 29.131 | 59.747 |
| DPKD | 41.354 | 64.890 | 06.388 | 49.542 | 37.986 | 29.253 | 60.063 |
| GKD | 41.191 | 64.770 | 06.670 | 49.070 | 37.900 | 28.274 | 60.460 |
| DCKD | 41.328 | **65.110** | 05.990 | 49.340 | 37.850 | 29.540 | 60.140 |
| ADPA | 40.288 | 60.185 | **06.670** | 48.626 | 37.450 | 28.760 | 60.039 |
| **TVKD (Ours)** | **41.605** | 64.550 | 05.760 | 49.650 | **38.540** | **29.880** | **61.250** |

**Detailed baseline setting**  For ADPA, following the original implementation [9], we first generated a single student rollout over the entire dataset. We then computed and saved the differences between the DPO teacher and the SFT teacher, which were used during training. For other online-based KD method(SeqKD,GKD), we use only single student or teacher rollout over the entire datasets following [9]. For online-based preference distillation method, we use ground-truth chosen and rejected answer instead of teacher and student rollouts following [9]. It can be different original implementation in theoretically. In the case of our method, we saved the DPO teacher's logits over the original dataset once and used them for training.

**Computational Efficiency**  We conducted our experiments using four Nvidia RTX 4090 GPUs. The total wall time for TVKD was approximately 2.5 to 3 hours. In our method, we precompute and store the DPO teacher's logits over the entire dataset once, which reduces computational overhead during training at the cost of increased storage usage. In contrast, methods such as GKD, SeqKD, and ADPA require generating additional student or teacher rollouts across the entire dataset and subsequently computing and storing teacher logits for each of those rollouts. This results in a larger storage overhead and slightly higher computational cost compared to our approach. In particular, ADPA requires not only the DPO-trained teacher but also the SFT-trained model to compute the logit differences, further increasing computational resource demands.

# F    Additional Results

## F.1    Open LLM Leaderboard

We show the overall score of the OLLs in the main table as shown below in  Table 6, Table 7.

## F.2    MT-bench

We represent the hexagons in  Figure 3 that represent the performance of each task on MT-bench. Our method shows consistently high performance compared to other methods, except for coding.

## F.3    Robustness to Distillation Strength

In  Figure 4a, we conduct an ablation study on the hyperparameter $\alpha$, which controls the influence of the teacher's value signal during training. We observe that TVKD achieves optimal performance on both the RM score and MT-bench when $\alpha = 0.7$, with performance dropping at both lower and higher values. This indicates that our method offers a tunable mechanism to balance teacher guidance and data-driven learning. A small $\alpha$ may underutilize the teacher's future preference signal, while an excessively large $\alpha$ can suppress the student's capacity to fit dataset-specific patterns. The ability

Table 7: Overall performance of Open LLM Leaderboard trained with Helpsteer2 datasets.

| Method | Average | ARC-easy | GSM8K | Hellaswag | MMLU | TruthfulQA | Winogrande |
|---|---|---|---|---|---|---|---|
| DPO teacher | 57.949 | 78.746 | 46.745 | 61.085 | 60.109 | 30.290 | 70.718 |
| SFT teacher | 53.152 | 76.200 | 47.300 | 60.844 | 60.070 | 04.500 | 70.000 |
| SFT | 41.064 | 64.983 | 06.369 | 48.676 | 37.787 | 29.376 | 59.195 |
| DPO | 41.243 | 64.680 | 05.980 | **49.572** | 37.972 | 29.253 | 60.000 |
| SimPO | 41.308 | 64.680 | 06.141 | 48.974 | 37.958 | 30.110 | 59.984 |
| WPO | 41.190 | 64.915 | 06.141 | 48.645 | 38.043 | 28.860 | 60.537 |
| TDPO | 41.130 | 64.140 | 07.126 | 49.114 | 36.767 | 28.886 | 60.805 |
| VanillaKD | 41.089 | 65.194 | 06.065 | 48.645 | 37.081 | 29.009 | 60.537 |
| SeqKD | 41.169 | 65.194 | 05.838 | 48.964 | **38.200** | 29.009 | 59.809 |
| DDPO | 41.234 | 65.109 | 06.293 | 49.522 | 37.958 | 29.009 | 59.511 |
| DPKD | 41.136 | 64.948 | **06.358** | 49.422 | 37.892 | 28.764 | 59.433 |
| GKD | 40.924 | 65.025 | 05.075 | 49.104 | 37.858 | 29.131 | 59.353 |
| DCKD | 41.092 | 65.125 | 06.520 | 48.516 | 37.253 | 28.519 | **60.616** |
| ADPA | 41.259 | **65.446** | 05.898 | 49.104 | 38.007 | 28.641 | 60.458 |
| **TVKD (Ours)** | **41.352** | 64.229 | 05.681 | 48.974 | 38.157 | **30.772** | 60.300 |

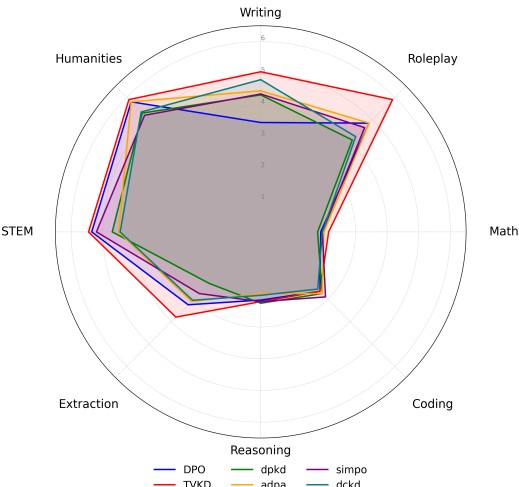

Figure 3: Visualization of token-wise preference

to adjust $\alpha$ provides practical flexibility, enabling users to calibrate the distillation strength to suit various datasets, teacher qualities, or deployment constraints.

### F.4 Robustness to Temperature Parameter

We present the robustness evaluation with respect to the temperature parameter $\beta$ in Figure 4b. Our method achieves the highest performance at $\beta = 2$, while overall demonstrating robust behavior across a wide range of values. This suggests that although higher temperatures yield a sharper value function and thus increase discriminability, the influence of $\beta$ remains limited since our method captures not the absolute value itself, but the difference in value before and after the current action.

### F.5 Teacher Model Analysis

We show additional experiments in Table 8 and Table 9 to demonstrate that TPKD is robust distillation to teacher quality. We compare the distillation performance on three different teachers: the default teacher model SFTed on Deita-10k-V0, a teacher model trained with Helpsteer2 as a DPO, and Instruct teacher, the instruct tuning version posted on Huggingface. Our method outperforms DCKD on all teacher settings. This shows that TPKD is relatively robust to the quality of the teachers. Note that Instruct teacher performs poorly on RM using helpsteer2's test set, but performs as well as DPO teacher on MT-bench, which is a comprehensive evaluation.

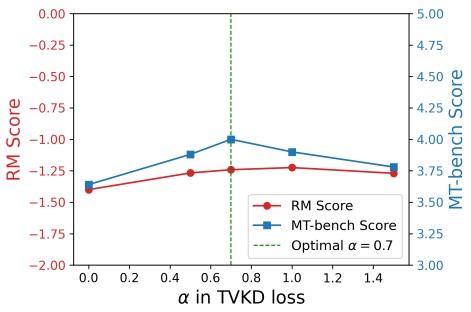

(a) Robustness to Distillation Strength

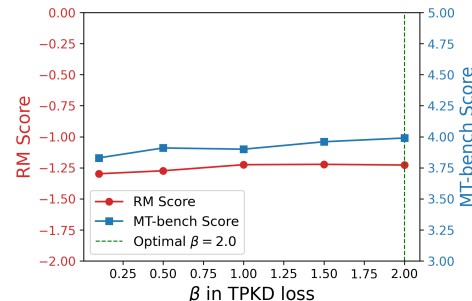

(b) Robustness to Temperature Parameter

Figure 4: Sensitivity analysis of distillation hyperparameters.

Table 8: RM (higher is better) comparison between DCKD and TPKD across teacher types.

| Teacher Type | DPO | SFT | Instruct |
|---|---|---|---|
| DCKD | -1.46 | -1.70 | -1.68 |
| TPKD | -1.21 | -1.46 | -1.56 |

Table 9: MT-bench (higher is better) comparison between DCKD and TPKD across teacher types.

| Teacher Type | DPO | SFT | Instruct |
|---|---|---|---|
| DCKD | 3.51 | 3.38 | 3.60 |
| TPKD | 3.98 | 3.73 | 3.96 |

