# OpenReview forum: "Preference Distillation via Value based Reinforcement Learning"
_NeurIPS.cc/2025/Conference — NeurIPS 2025 poster_

### Official Review · Reviewer_5RhB · 2025-06-29

**Clarity:** 3
**Significance:** 2
**Originality:** 3
**Rating:** 4
**Confidence:** 3

**Summary:**

The paper introduces Teacher Value-based Knowledge Distillation (TVKD), a novel method for aligning small language models with human preferences by leveraging the value function of a larger, DPO-aligned teacher model. Unlike traditional approaches that rely on coarse binary feedback or behavior cloning, TVKD uses the teacher's value function to provide fine-grained, sequence-level reward signals via Potential-based Reward Shaping. It is a fully offline method that requires no additional rollouts. Experiments demonstrate TVKD's superiority over baselines across multiple alignment benchmarks like MT-Bench and AlpacaEval.

**Questions:**

The use of a value function as a shaping reward introduces a risk of reward hacking, where the student learns to exploit artifacts in the teacher’s value estimates rather than genuinely improving preference alignment. What are authors' opinions on this aspect?

**Ethical Concerns:**

["NO or VERY MINOR ethics concerns only"]

**Final Justification:**

I have read the responses from authors and they address my concerns. I will keep my score.

**Limitations:**

yes

**Quality:**

3

**Strengths And Weaknesses:**

Strength:
1. Authors use the teacher model's value function as a soft reward signal, effectively address the coarse binary feedback limitation of standard DPO. They frame this as a Potential-based Reward Shaping (PBRS) problem, which bypass the need for online rollouts.
2. Authors provide theoretical proofs and analysis on the method, which establishes strong theoretical guarantees.
3. Experimental results show the effectiveness of this method on several models and benchmarks.

Weakness:

1. The paper could benefit from more discussion of the relationship between the hyperparameter $\beta$ in DPO and the distillation strength $\alpha$, as these parameters likely interact in important ways.
2. While the paper highlights the advantage of being an offline method compared to online approaches like ADPA, it does not quantify the computational overhead TVKD introduces over standard DPO. It would be helpful to include a direct comparison on training time and resource usage.
3. Compared to MT-Bench and Alpaca Eval, Arena Hard may be a more suitable benchmark nowadays.

---

> ### Author Rebuttal · Authors · 2025-07-31
>
> We would like to thank reviewer 5RhB for their constructive feedback and for highlighting the value of our theoretical and experimental contributions.
>
> ### **W1.**
>
> ---
>
> > The paper could benefit from more discussion of the relationship between the hyperparameter $\beta$ in DPO and the distillation strength $\alpha$, as these parameters likely interact in important ways.
> >
>
> ### **R1.**
>
> ---
>
> As shown in Equation (3), **β controls the sensitivity to preference margins between the winning and losing responses**, while **α determines the strength of the teacher’s auxiliary reward signal**.
>
> We further explore their interaction in Appendix F.4 through a series of sweep experiments. The results show that **TVKD remains relatively robust across a wide range of α and β values**.
>
> Jointly tuning both parameters can be computationally expensive. To address this, we recommend first selecting β using standard DPO without distillation, and then adjusting α with β fixed. This strategy is motivated by the observation that β has a greater impact on the overall reward scale, whereas α primarily modulates the influence of the teacher signal. For each β, it is possible to find a suitable α that ensures effective learning. We will include this recommendation in the camera-ready version.
>
> ### **W2.**
>
> ---
>
> > While the paper highlights the advantage of being an offline method compared to online approaches like ADPA, it does not quantify the computational overhead TVKD introduces over standard DPO. It would be helpful to include a direct comparison on training time and resource usage.
> >
>
> ### **R2.**
>
> ---
>
> As shown in Appendix E, TVKD requires **only a single additional forward pass** of the teacher model over the entire dataset compared to standard DPO. This introduces a minor overhead relative to DPO, but the **cost of a one-time forward computation is relatively small.** Moreover, this additional cost is justified by the performance improvements achieved through distillation.
>
> When compared to **common distillation baselines such as GKD and SeqKD, TVKD incurs the same computational cost**, as all these methods require precomputing teacher outputs once. We also report a table comparing the **CPU wall time** of standard DPO and other methods. As the table shows, the time difference between DPO and TVKD is less than 25 minutes, and GKD and TVKD take almost the same time. In contrast, DCKD and ADPA incur significantly higher computational costs.
>
> | Method | CPU wall time |
> | --- | --- |
> | DPO | **3h 20m** |
> | GKD | 3h 42m |
> | DCKD | 4h 22m |
> | ADPA | 5h 2m |
> | TVKD(Ours) | 3h 41m |
>
> ### **W3.**
>
> ---
>
> > Compared to MT-Bench and Alpaca Eval, Arena Hard may be a more suitable benchmark nowadays.
> >
>
> ### **R3.**
>
> ---
>
> | Method | Arena-Hard-Auto Win Rate(%) |
> | --- | --- |
> | **TVKD(Ours)** | **54.4%** |
> | ADPA | 50.0% |
> | DPO | 34.8% |
>
> We appreciate the reviewer’s suggestion and agree that Arena-Hard is an increasingly relevant benchmark. We report below the win rates of TVKD, ADPA, and DPO on the Arena-Hard-Auto [1] benchmark, where TVKD continues to demonstrate a performance advantage.
>
> We will make an effort to include as many Arena-Hard results as possible in the final camera-ready version of the paper to further strengthen our empirical evaluation.
>
> [1] https://github.com/lmarena/arena-hard-auto
>
> ### **Q1.**
>
> ---
>
> > The use of a value function as a shaping reward introduces a risk of reward hacking, where the student learns to exploit artifacts in the teacher’s value estimates rather than genuinely improving preference alignment. What are authors' opinions on this aspect?
> >
>
> ### **A1.**
>
> ---
>
> Simply using the teacher value function directly for distillation can lead to reward hacking. However, TVKD mitigates this risk by incorporating the teacher value function into the student reward term in a way that satisfies the PBRS condition. PBRS theoretically guarantees that the optimal policy remains the same as when using only the offline DPO dataset. As shown in Table 5, performance degrades when using similar terms that do not satisfy the PBRS condition, which provides empirical evidence of the importance of PBRS. This is one of the key contributions of TVKD, as it demonstrates a safe and effective way to leverage the teacher value function.

---

> ### Comment · Area_Chair_zAgg · 2025-08-04
>
> Dear Reviewer,
>
> Please engage in the discussion with the authors. The discussion period will end in a few days.
>
> Thanks,
>
> AC

---

### Official Review · Reviewer_oStu · 2025-06-30

**Clarity:** 3
**Significance:** 3
**Originality:** 3
**Rating:** 5
**Confidence:** 4

**Summary:**

The paper introduces Teacher Value-based Knowledge Distillation (TVKD) to enhance preference alignment in small language models by leveraging the value function of large language models. Direct Preference Optimization (DPO) relies on coarse-grained binary feedback, which is insufficient for small models with limited capacity. Existing distillation methods (e.g., KL divergence) mimic teacher behavior but overlook reward modeling, risking misalignment with human preferences. TVKD introduces potential-based reward shaping (PBRS) by treating the teacher model’s value function as an auxiliary reward signal, preserving DPO’s optimal policy while providing fine-grained guidance. Experiments across benchmarks MT-Bench, AlpacaEval, Open LLM Leaderboard show consistent gains over baselines.

**Questions:**

Question:
* How can the potential for transferring spurious biases from the teacher model be further quantified or mitigated? Have alternative methods been explored, such as teacher ensembles or teachers trained with more robust alignment techniques?
* Beyond current benchmarks, how does TVKD's generalization hold on highly out-of-domain or adversarial datasets to truly assess its robustness against reward hacking?
* A more detailed analysis of the hyperparameter α's sensitivity is needed. How does its tuning affect the balance between teacher guidance and student learning from direct DPO, and what are practical tuning guidelines?
* Can the TVKD framework be generalized or adapted to distill from teachers aligned using other RLHF methods, such as those employing an explicit reward model (e.g., PPO-trained teachers)?

**Ethical Concerns:**

["NO or VERY MINOR ethics concerns only"]

**Final Justification:**

The author's response address my concern on the optimality requirement of the teacher's value, with additional results and further clarification. Thus I decide to raise my score.

**Limitations:**

yes

**Quality:**

3

**Strengths And Weaknesses:**

Strengths:
* Theoretically sound: Leverages token-level optimal value of soft value-based RL (Lemma 1), with an invariance guarantee of the optimal policy under PBRS (Lemma 2), avoiding issues of action-dependent shaping (Corollary 2.1).
* Empirical performance: Comprehensive experiments on diverse model sizes (Table 2) and datasets (DPO-MIX-7K, HelpSteer2) demonstrate TVKD’s superiority over 14 baselines across RM, MT-Bench, AlpacaEval, and Open LLM Leaderboard metrics.

Weaknesses:
* reward hacking risk from teacher bias: Distillation from a DPO-trained teacher, whose "optimal" policy might contain spurious biases from preference data, increases the risk of transferring these biases and impairing student generalization on out-of-domain datasets.
* teacher model "optimality" assumption: The "optimality" of the teacher's value function is derived from its training on potentially imperfect human preference data, which might not perfectly reflect true human preferences, thus limiting the distilled signal.

---

> ### Author Rebuttal · Authors · 2025-07-31
>
> We appreciate reviewer oStu’s thoughtful comments and are grateful for the recognition of both our theoretical analysis and experimental validation.
>
> ### **W1.**
>
> ---
>
> > reward hacking risk from teacher bias: Distillation from a DPO-trained teacher, whose "optimal" policy might contain spurious biases from preference data, increases the risk of transferring these biases and impairing student generalization on out-of-domain datasets.
> >
>
> ### **R1.**
>
> ---
>
> We acknowledge the reviewer’s concern regarding reward hacking risks arising from teacher bias in DPO-based distillation.
>
> We identify **two sources of reward hacking** in this context. The first stems from the sparsity of **binary preference labels in the DPO dataset** combined with the **limited capacity of the student** model, which may lead the student to overfit **misleading signals**. Our distillation framework directly addresses this by **leveraging richer supervision from the teacher model**.
>
> The second source arises from the potential **biases encoded in the teacher model** itself, which can be **transferred to the student**. To mitigate this, we introduce an **auxiliary reward term that satisfies the PBRS condition**, ensuring that the student learns a policy **aligned with the original DPO objective**. This theoretically guarantees the preservation of the optimal policy and protects against harmful bias propagation. Supporting evidence can be found in Table 5, where PBRS-compliant (state-dependent) shaping leads to higher margin accuracy.
>
> As for out-of-domain generalization, **Open LLM leaderboard(OLL) in Table 1** approximates broad language abilities that contain largely disjoint from the training dataset. TVKD outperforms strong baselines on OLL as well, suggesting enhanced robustness in generalization compared to other methods.
>
> ### **W2.**
>
> ---
>
> > teacher model "optimality" assumption: The "optimality" of the teacher's value function is derived from its training on potentially imperfect human preference data, which might not perfectly reflect true human preferences, thus limiting the distilled signal.
> >
>
> ### **R2.**
>
> ---
>
> We appreciate the concern, but TVKD does not assume that the teacher’s value function is optimal. Instead, it uses the teacher signal solely as a potential-based shaping term, which theoretically preserves the optimal policy of the original DPO objective (Lemma 2).
>
> This design ensures that the dataset-derived preference signal remains the primary supervision, and the teacher value merely provides auxiliary guidance. As shown in Appendix F.5, **TVKD shows greater robustness than baseline methods even when the teacher is trained suboptimally**, confirming that our approach does not rely on the optimality of the teacher model.
>
> ### **Q1.**
>
> ---
>
> > How can the potential for transferring spurious biases from the teacher model be further quantified or mitigated? Have alternative methods been explored, such as teacher ensembles or teachers trained with more robust alignment techniques?
> >
>
> ### **A1.**
>
> ---
>
> To approximate the effect of transferring teacher bias, we evaluated how much the **original DPO margin accuracy(Margin Acc.) degrades after distillation** in Table 5. This serves as an indirect indicator of how strongly the student is influenced by teacher information. Through experiments with various distillation terms, we observed that **terms satisfying PBRS(State dependent) tend to follow the original DPO dataset’s preference signal more faithfully**. This suggests that the original preference signal in the DPO dataset is more strongly preserved than the potential biases of the teacher, indicating that the student model remains primarily aligned with the dataset rather than being dominated by teacher bias.
>
> ### **Q2.**
>
> ---
>
> > Beyond current benchmarks, how does TVKD's generalization hold on highly out-of-domain or adversarial datasets to truly assess its robustness against reward hacking?
> >
>
> ### **A2.**
>
> ---
>
> Theoretically, TVKD is designed to follow the preference structure encoded in the DPO dataset, regardless of the teacher’s performance, thereby aiming to **preserve the original reward signal**. However, due to the inherently unstable training dynamics of reinforcement learning, incorrect auxiliary rewards can lead to slower convergence or suboptimal local minima.
>
> The case suggested by the reviewer could occur in a hypothetical scenario where a teacher is trained to intentionally mislead students. Although this setting differs from a typical distillation setting and we cannot be certain, we predict that TVKD does not guarantee success in this scenario.
>
> ### **Q3.**
>
> ---
>
> > A more detailed analysis of the hyperparameter α's sensitivity is needed. How does its tuning affect the balance between teacher guidance and student learning from direct DPO, and what are practical tuning guidelines?
> >
>
> ### **A3.**
>
> ---
>
> We present a sweep over **α** in Appendix F.3 and Figure 4. As expected, **a large α makes the student overly follow the teacher’s value signal, weakening the DPO objective, while a very small α reduces TVKD to standard offline DPO.** Within the range of **0.1–1.5**, we observed stable performance gains without significant drops, but we cannot rule out performance degradation for more extreme α values. As a practical guideline, we suggest starting with **α ≈ 0.5** and adjusting within the tested range.
>
> ### **Q4.**
>
> ---
>
> > Can the TVKD framework be generalized or adapted to distill from teachers aligned using other RLHF methods, such as those employing an explicit reward model (e.g., PPO-trained teachers)?
> >
>
> ### **A4.**
>
> ---
>
> In principle, **TVKD can be naturally extended to teacher models trained with explicit Q-functions**, such as PPO or Soft Actor-Critic (SAC), because these methods inherently provide a value function. To our knowledge, however, **no prior work has explored distillation across fundamentally different RLHF frameworks (e.g., DPO, PPO, SAC) simultaneously**, largely due to their reliance on distinct data structures. Such an experiment would require a dataset where each query has both pairwise preference comparisons and scalar rewards.
>
> In such scenarios, we are not certain whether a DPO-trained or PPO-trained teacher would serve as a better source for distillation, as the outcome would likely depend heavily on the characteristics of the dataset, such as label format, reward signal fidelity, and distributional coverage. We believe this opens up an interesting avenue for future research into how different RLHF paradigms interact with distillation dynamics.

---

> > ### Comment · Reviewer_oStu · 2025-08-05
> >
> > Thanks for the author's further clarification. After re-examination of the results in the manuscript, I acknowledge that TVKD does not rely on optimal teacher value, which addresses my major concern. Thus I decide to raise my score. I have some further questions. Can the author provide an intuition of what useful signal is distilled into the student when the teacher is sub-optimal? Furthermore, how does the "sub-optimality" of the teacher affect the distillation process, e.g., when the teacher is merely trained by DPO, is this distillation process still beneficial?

---

> > > ### Author Response · Authors · 2025-08-06
> > >
> > > Thank you for your thoughtful review. We are glad that our contribution seems to have come across well.
> > > Below are answers to your additional questions:
> > > Intuitively, Teacher's signal behaves like a soft label for a DPO dataset. The DPO dataset has two binary labels, ‘preferred’ or ‘not preferred’, and Teacher passes a value for ‘how much more preferred’ as an auxiliary reward.
> > > However, this can lead to reward conflicts if the value fails to satisfy the PBRS, such as the product of Teacher's log-prob. TVKD presents a theoretical method to resolve this and demonstrates it experimentally.
> > > Even with sub-optimal information from the teacher, the teacher has a relative advantage over the student in extracting the “nuances” of the preferred answer from the binary label of the DPO. This is because teachers are typically larger, better trained, and have higher language ability. Passing this information to the student can provide better soft labels than using the DPO dataset alone.

---

> ### Comment · Area_Chair_zAgg · 2025-08-04
>
> Dear Reviewer,
>
> Please engage in the discussion with the authors. The discussion period will end in a few days.
>
> Thanks,
>
> AC

---

### Official Review · Reviewer_uzh2 · 2025-07-01

**Clarity:** 3
**Significance:** 2
**Originality:** 2
**Rating:** 4
**Confidence:** 3

**Summary:**

This paper introduces Teacher Value-based Knowledge Distillation (TVKD), a method to distill preference alignment from a teacher model. Instead of focusing solely on behavior cloning or KL divergence, TVKD incorporates a value-based auxiliary reward that adheres to potential-based reward shaping principles, preserving the global reward structure and optimal policy. The approach integrates seamlessly into standard DPO training without requiring additional rollouts. Experimental results show that TVKD yields consistent performance gains across multiple benchmarks and model sizes.

**Questions:**

Which theoretical results or proofs are original contributions by the authors, rather than adaptations or direct reuse from prior work?

**Ethical Concerns:**

["NO or VERY MINOR ethics concerns only"]

**Final Justification:**

Thanks for authors to address my concerns, I will increase my score.

**Limitations:**

Yes

**Quality:**

3

**Strengths And Weaknesses:**

**Strengths**
1. The authors provide detailed theoretical analyses supporting the design of their method, which enhances its credibility and soundness.
2. The paper includes comprehensive experiments with a wide range of baseline comparisons, demonstrating thorough empirical evaluation.

**Weakness**
1. According to the Qwen3 Technical Report, SOTA models typically undergo multiple preference optimization stages beyond DPO alone. This raises concerns about whether a DPO-based distillation method can sufficiently capture the full spectrum of preference knowledge from a powerful teacher model.

2. The proposed reward shaping technique using the teacher's value function appears similar to Equation 10 in the paper “From r to Q*: Your Language Model is Secretly a Q-Function.” The authors should clarify the conceptual and practical differences between their approach and that prior work to avoid ambiguity and better position their contribution.

3. Experimentally, the proposed method does not outperform the ADAP baseline by a significant margin. Given that evaluation relies on LLM-based judges—known to exhibit randomness and occasional misalignment with true human preferences—it is unclear whether the observed improvements are robust.

---

> ### Author Rebuttal · Authors · 2025-07-31
>
> We sincerely thank reviewer uzh2 for acknowledging the theoretical contributions and extensive experiments in our work, and we appreciate the thoughtful and constructive feedback provided.
>
> ### **W1.**
>
> ---
>
> > According to the Qwen3 Technical Report, SOTA models typically undergo multiple preference optimization stages beyond DPO alone. This raises concerns about whether a DPO-based distillation method can sufficiently capture the full spectrum of preference knowledge from a powerful teacher model.
> >
>
> ### **R1.**
>
> ---
>
> **In principle, TVKD is compatible with any teacher model trained under the Maximum Entropy RL framework(MaxEnt RL).** While the Qwen3 Technical Report does not provide full algorithmic details of all training stages, we have confirmed that GRPO is explicitly mentioned in their pipeline. Beyond GRPO, methods like PPO and SAC also fall under the MaxEnt RL family and are theoretically compatible with TVKD via the soft value function formulation in Lemma 1.
>
> Moreover, **when the teacher model provides an explicit value function (some variants of PPO and GRPO) TVKD can directly leverage it without relying on Lemma 1**, further simplifying implementation.
>
> Although our experiments focus on DPO-trained teachers, this choice was made to ensure methodological consistency between teacher and student models. **TVKD itself is not limited to DPO teachers** and can, in principle, be extended to a broader range of RLHF-trained models.
>
> If there are additional preference optimization methods used in Qwen3 or elsewhere that we have not accounted for, please join the discussion and let us know. **We would be happy to discuss their compatibility with TVKD**. We appreciate the reviewer’s perspective and welcome further dialogue on this point.
>
> ### **W2.**
>
> ---
>
> > The proposed reward shaping technique using the teacher's value function appears similar to Equation 10 in the paper “From r to Q*: Your Language Model is Secretly a Q-Function.” The authors should clarify the conceptual and practical differences between their approach and that prior work to avoid ambiguity and better position their contribution.
> >
>
> ### **R2.**
>
> ---
>
> We appreciate the reviewer’s comment and agree that clarifying the relationship with *“From r to Q: Your Language Model is Secretly a Q-Function” (rtoQ)* is important.
>
> First, we would like to emphasize that the primary goal and contribution of our work is to **utilize teacher information** in a way that does not interfere with the reward structure of the **offline DPO dataset during distillation**. In contrast, rtoQ is primarily an **analytical work** that reinterprets DPO from a Q-learning perspective and does not propose a new training method or distillation technique.
>
> Our training loss may looks similar to Equation 10 in rtoQ because **both share the PBRS condition** (originally from Ng et al., ICML 1999 [1]). However, while rtoQ employs this equation solely for **theoretical analysis**, TVKD explicitly introduces the teacher value function $V_\phi$ as a shaping term to **directly guide student training**.
> **PBRS is a widely used theoretical tool in RL** ([2],[3],[4]), and many reward shaping methods have been developed based on this principle. Our theoretical analysis and method are also built upon this foundation, leveraging PBRS to provide a principled understanding of the distillation process.
>
> [1] Ng, Andrew Y., Daishi Harada, and Stuart Russell. "Policy invariance under reward transformations: Theory and application to reward shaping." *Icml*. Vol. 99. 1999.
>
> [2] Wiewiora, Eric, Garrison W. Cottrell, and Charles Elkan. "Principled methods for advising reinforcement learning agents." Proceedings of the 20th international conference on machine learning (ICML-03). 2003.
>
> [3] Asmuth, John, Michael L. Littman, and Robert Zinkov. "Potential-based Shaping in Model-based Reinforcement Learning." AAAI. 2008.
>
> [4] Gao, Yang, and Francesca Toni. "Potential Based Reward Shaping for Hierarchical Reinforcement Learning." IJCAI. 2015.
>
> ---
>
> ### **W3.**
>
> ---
>
> > Experimentally, the proposed method does not outperform the ADAP baseline by a significant margin. Given that evaluation relies on LLM-based judges—known to exhibit randomness and occasional misalignment with true human preferences—it is unclear whether the observed improvements are robust.
> >
>
> ### **R3.**
>
> ---
>
> We acknowledge that LLM-based judges can sometimes be misaligned with true human preferences, so we aimed to mitigate this by employing a diverse set of evaluation methods. Specifically, we used a **reproducible open-source reward model (RM), GPT-judge-based MT-bench(MT), 1-on-1 pairwise comparisons (AE) with ADPA, and the Open LLM Leaderboard (OLL)**, which evaluates general commonsense reasoning beyond the training set. **As shown in Table 1, TVKD consistently outperforms ADPA across all these evaluations.**
>
> To ensure statistical reliability, we report the standard deviations from three independent trainings of the experiments in Table 1. Due to computational cost, we only provide standard deviation values for the comparison of ADPA and TVKD. **TVKD outperforms ADPA with statistical significance on all benchmarks.**
>
> | DPOMIX | RM&nbsp;($\uparrow$) | MT&nbsp;($\uparrow$) | AE&nbsp;($\uparrow$) | OLL&nbsp;($\uparrow$) |
> |--------|---------------------|----------------------|----------------------|-----------------------|
> | ADPA   | -1.2418 ($\pm$ 0.037) | 3.7445 ($\pm$ 0.056) | 50.00 | 40.2883 ($\pm$ 0.0394) |
> | Ours   | **-1.1327** ($\pm$ 0.033) | **3.9864** ($\pm$ 0.087) | **53.09** ($\pm$ 0.127) | **41.6237** ($\pm$ 0.0465) |
>
> Additionally, we emphasize that **TPKD does not require online rollouts**. ADPA evaluates student outputs with the teacher at every epoch, resulting in increased storage and computational overhead as both the dataset and teacher model size grow. In contrast, TPKD requires only a single computation of the teacher value function over the entire dataset, making it significantly more computationally efficient. **A detailed analysis of computation efficiency can be found in Appendix E.1.**
>
> And we also report CPU wall time of comparable methods. Our method demonstrates a shorter wall time than ADPA, despite achieving higher performance in all benchmarks.
>
> | Method | CPU wall Time |
> | --- | --- |
> | DPO | **3h 20m** |
> | GKD | 3h 42m |
> | DCKD | 4h 22m |
> | ADPA | 5h 2m |
> | TVKD(Ours) | 3h 41m |
>
> ### **Q1.**
>
> ---
>
> > Which theoretical results or proofs are original contributions by the authors, rather than adaptations or direct reuse from prior work?
> >
>
> ### **A1.**
>
> ---
>
> We appreciate the opportunity to further clarify our original contributions.
>
> We introduce TVKD, a novel method for **incorporating the rich features of a teacher model** while **minimizing interference with the reward structure of the offline dataset**. We also provide a theoretical guarantee in **Lemma 2**, proving that TVKD preserves the optimal policy under the PBRS condition. In addition, we present **extensive empirical results** demonstrating that TVKD consistently outperforms strong baselines across various settings, highlighting its robustness and effectiveness.
>
> As our work builds upon the theoretical foundations of reinforcement learning, some proof techniques and equation forms may naturally resemble those used in prior work. However, identifying limitations in DPO-based distillation through theoretical analysis, proposing a new method to address them, and empirically validating its effectiveness are clear contributions of our work.
>
> We believe this contribution is original and offers a new perspective on DPO-based distillation, both theoretically and practically, which we hope will be valuable to the community.

---

> > ### Comment · Reviewer_uzh2 · 2025-08-04
> >
> > Thanks for authors to address my concerns, I will increase my score.

---

> ### Comment · Area_Chair_zAgg · 2025-08-04
>
> Dear Reviewer,
>
> Please engage in the discussion with the authors. The discussion period will end in a few days.
>
> Thanks,
>
> AC

---

### Official Review · Reviewer_Sqwj · 2025-07-03

**Clarity:** 3
**Significance:** 3
**Originality:** 3
**Rating:** 3
**Confidence:** 4

**Summary:**

This paper addresses the challenge of preference alignment for small language models (SLMs) by proposing a novel knowledge distillation approach termed Teacher Value-based Knowledge Distillation (TVDK).

While current mainstream methods such as Direct Preference Optimization (DPO) align models using binary comparisons, the resulting coarse-grained signals may be insufficient for small models with limited capacity. Traditional knowledge distillation approaches primarily focus on mimicking the behavior of the teacher model, neglecting the richer internal reward modeling contained within it.

The core idea of TVKD lies in moving beyond mere imitation of the teacher’s output, instead distilling the teacher’s internal "value judgments." This is achieved by introducing an auxiliary reward term that satisfies the theory of Potential-Based Reward Shaping (PBRS). As a result, the student model receives more fine-grained guidance while the original optimization objective and optimal policy of DPO remain unchanged.

**Questions:**

1. On what data was the teacher model trained using DPO?
The manuscript mentions that the teacher model πϕ is one that has "already been trained using DPO." But the specifics of this training process are quite vague.
2. Definition and Interpretation of "Accuracy" in Value Function Analysis (Table 3).
In the analysis presented in Section 4.3, Table 3 showcases a metric named "Accuracy(%)", which is utilized to demonstrate that its shaping term can better identify preferred trajectories. Nevertheless, both the implication and the results of this metric are perplexing.The accuracy of the "Ours" method indicated in the paper is 25.82%. This figure, on its own, appears remarkably low, making it difficult from an intuitive standpoint to regard as "accurate".

**Ethical Concerns:**

["NO or VERY MINOR ethics concerns only"]

**Final Justification:**

The manuscript and subsequent rebuttal present detailed experiments and results, and the authors addressed some of my concerns. However, I remain concerned about the strong reliance on the teacher model and the associated biases. I look forward to future work. This is my final rating.

**Limitations:**

Yes

**Quality:**

3

**Strengths And Weaknesses:**

The method's principal advantage is its theoretically-grounded framework that distills a teacher model's internal value function via potential-based reward shaping, thereby enriching the student's learning with fine-grained supervision while provably preserving the optimal policy. Conversely, its core limitation is the fundamental dependence on the teacher model's quality, as any inaccuracies or biases inherent in the teacher's approximated value function are directly propagated to the student, capping performance and potentially transferring undesirable behaviors.

---

> ### Author Rebuttal · Authors · 2025-07-31
>
> We sincerely appreciate the reviewer Sqwj’s recognition of our theoretical analyses and comprehensive experiments, and we are grateful for the constructive comments provided. Below, we respond to the specific points raised:
>
> ### **W1,2.**
>
> ---
>
> > Conversely, its core limitation is the fundamental dependence on the teacher model's quality, as any inaccuracies or biases inherent in the teacher's approximated value function are directly propagated to the student, capping performance and potentially transferring undesirable behaviors.
> >
>
> ### **R1.**
>
> ---
>
> We understand the concern that errors or biases in the teacher’s value function may be propagated to the student, which is a well-known issue in distillation methods. To address this issue, TVKD is designed **to utilize the rich representations of the teacher** model while explicitly **preserving the human-preference structure** encoded in the dataset, as described in lines 36–37 of the Introduction.
>
> **Lemma 2 provides the theoretical foundation** for this design. By incorporating the teacher’s value function as a potential within the PBRS framework, TVKD preserves the optimal policy of standard DPO and reduces the influence of bias present in the teacher. **This is supported by empirical results in Table 5**, where the state-dependent variant that satisfies the PBRS condition achieves higher DPO-margin accuracy than the action-dependent variant that violates it.
>
> Additionally, **Appendix F.5 compares distillation using teachers before and after DPO fine-tuning**. When the teacher model is of lower quality, such as before undergoing DPO, TVKD remains more **robust than baseline methods**. This shows that TVKD is more **robust to low-quality teachers** by prioritizing supervision signals in the dataset over teacher guidance.
>
> Taken together, these theoretical and empirical results demonstrate that TVKD can effectively leverage the benefits of the teacher model while mitigating the risks of negative bias transfer.
>
> ---
>
> ### **Q1.**
>
> ---
>
> > On what data was the teacher model trained using DPO? The manuscript mentions that the teacher model $π_ϕ$ is one that has "already been trained using DPO." But the specifics of this training process are quite vague.
> >
>
> ### **A1.**
>
> ---
>
> Thank you for pointing out the lack of clarity in our description. As described in **Section 4.1 and Appendix Section E.1**, the DPO teacher model was first **supervised fine-tuned (SFT) on the Deita-10k-v0 dataset**, and then **further trained with DPO for 2 epochs on the same dataset as the student model (DPOMIX or HelpSteer2)**. This setup was chosen to **ensure fairness by following the same configuration as the ADPA paper**. The performance of the trained DPO teacher, which consistently surpasses both the student model and the SFT teacher, can be verified in **Table 1**. All training details can be verified in the provided codebase.
>
> ---
>
> ### **Q2.**
>
> ---
>
> > Definition and Interpretation of "Accuracy" in Value Function Analysis (Table 3). In the analysis presented in Section 4.3, Table 3 showcases a metric named "Accuracy(%)", which is utilized to demonstrate that its shaping term can better identify preferred trajectories. Nevertheless, both the implication and the results of this metric are perplexing.The accuracy of the "Ours" method indicated in the paper is 25.82%. This figure, on its own, appears remarkably low, making it difficult from an intuitive standpoint to regard as "accurate".
> >
>
> ### **A2.**
>
> ---
>
> We acknowledge that our explanation of Table 3 was insufficient and appreciate the reviewer’s thoughtful comment. **Table 3 reports results on the test split of DPOMIX, not on the training data.** The relatively low accuracy primarily reflects the inherent difficulty, rather than an issue with insufficient training.
>
> For clarity, we report below the accuracy on the **training split**, where all methods achieve **above 70%**, indicating that the models are well-trained. In both the training and test datasets, our method consistently demonstrates higher accuracy compared to the baselines.
>
> | Method | Log Prob | Log Prob with LC | Ours |
> | --- | --- | --- | --- |
> | Test Accuracy (%) | 19.56 | 19.72 | **25.82** |
> | Training Accuracy (%) | 75.54 | 77.49 | **83.81** |
>
> We promise to include this table in the Appendix and provide a clearer explanation in the camera-ready version.

---

> ### Comment · Area_Chair_zAgg · 2025-08-04
>
> Dear Reviewer,
>
> Please engage in the discussion with the authors. The discussion period will end in a few days.
>
> Thanks,
>
> AC

---

### Note · Authors · 2025-08-12

We sincerely appreciate the hard work of the AC and the reviewers who participated in the discussion.

All the reviewers pointed out the theoretically grounded framework of TVKD as an advantage. Our two lemmas provide theoretical support for TVKD. They also identified the improvement in performance on different model sizes and datasets as an advantage. We are grateful to the reviewers for recognising our contributions, both theoretical and empirical.

We also appreciate the constructive feedback from reviewers that allowed us to clarify our work during the rebuttal and discussion process. During the discussion, we believe that we addressed most of the concerns raised by the reviewers. While we believe that the main concerns raised by reviewer Sqwj were covered in our rebuttal, it is unfortunate that we did not have the opportunity to receive additional feedback from Sqwj during the discussion phase. We hope that this will be taken into account in the decision of AC.

We believe that the paper has both theoretical contributions and empirical analysis, which will contribute to the SLM alignment community.

Thank you

---

### Decision · Program_Chairs · 2025-09-17

**Decision:**

Accept (poster)

**Comment:**

This paper introduces a method to distill preference alignment from a teacher model, called Teacher Value-based Knowledge Distillation (TVKD). The latter incorporates a value-based auxiliary reward that adheres to principles from potential-based reward shaping to make sure that the global reward structure and optimal policy can be preserved. The approach integrates into standard DPO training without the need for additional rollouts. Experimental results, in which the authors compare to multiple benchmarks and model sizes, are promising.

Overall, the reviewers are quite positive about the paper. In their reports, they raised a couple of comments and concerns, however, these could essentially be resolved in the rebuttal and subsequent discussion.